# Hyperpruning: Efficient Pruning Through Lyapunov Metric Hypersearch

## Abstract

Various pruning methods have been introduced for over-parameterized recurrent neural networks to improve efficiency in terms of power and storage. With the advance in pruning methods and their variety, a new problem of 'hyperpruning' is becoming apparent: finding a suitable pruning method with optimal hyperparameter configuration for a particular task and network. Such search is different from the standard hyperparameter search, where the accuracy of the optimal configuration is unknown. In the context of network pruning, the accuracy of the non-pruned (dense) model sets the target for the accuracy of the pruned model. Thereby, the goal of hyperpruning is to reach or even surpass this target. It is critical to develop efficient strategies for hyperpruning since direct search through pruned variants would require time-consuming training without guarantees for improved performance. To address this problem, we introduce a novel distance based on Lyapunov Spectrum (LS) which provides means to compare pruned variants with the dense model and early in training to estimate the accuracy that pruned variants will achieve after extensive training. The ability to predict performance allows us to incorporate the LS-based distance with Bayesian hyperparameter optimization methods and to propose an efficient and first-of-its-kind hyperpruning approach called *LS*-based *H*yperpruning (***LSH***) which can optimize the search time by an order of magnitude compared to standard full training search with the loss (or perplexity) being the accuracy metric. Our experiments on stacked LSTM and RHN language models trained with the Penn Treebank dataset show that with a given budget of training epochs and desired pruning ratio, LSH obtains more optimal variants than standard loss-based hyperparameter optimization methods. Furthermore, as a result of the search, LSH identifies pruned variants that outperform state-of-the-art pruning methods and surpass the accuracy of the dense model.

## 1 Introduction

Over the last decade, the performance of sequence models, i.e. Recurrent Neural Networks (RNN), has been significantly enhanced in various applications such as action recognition (Su et al., 2020), video summarization (Zhao et al., 2018) and voice conversion (Huang et al., 2021). In particular, RNN variants such as LSTM (Hochreiter & Schmidhuber, 1997; Zaremba et al., 2014; Malhotra et al., 2015) and RHN (Zilly et al., 2017) excel in various NLP application ranging from machine translation (Wu et al., 2016) to language modeling (Irie et al., 2019). However, the architectural inherent computational demands of RNN, being linear to input sequence length and quadratic to model size, lead to a slowdown in training and inference. This hinders these models from being deployed on resource-limited devices such as mobile devices. Multiple methods have been proposed to alleviate this problem, including network quantization (Hernández et al., 2020; Han et al., 2015a) and weight sharing (Ullrich et al., 2017). Among these approaches, network pruning is advantageous since aims to achieve a sparse model which would require fewer computational resources. A particularly notable pruning approach is *dense-to-sparse*, where the network is gradually being pruned starting from a non-pruned (dense) model (Han et al., 2015b; Guo et al., 2016). While the inference time of the pruned network will eventually decrease, the training time remains similar to or even longer than the time of training a dense model. Such training time typically extends to multiple days or weeks and poses challenges in achieving pruned models efficiently.

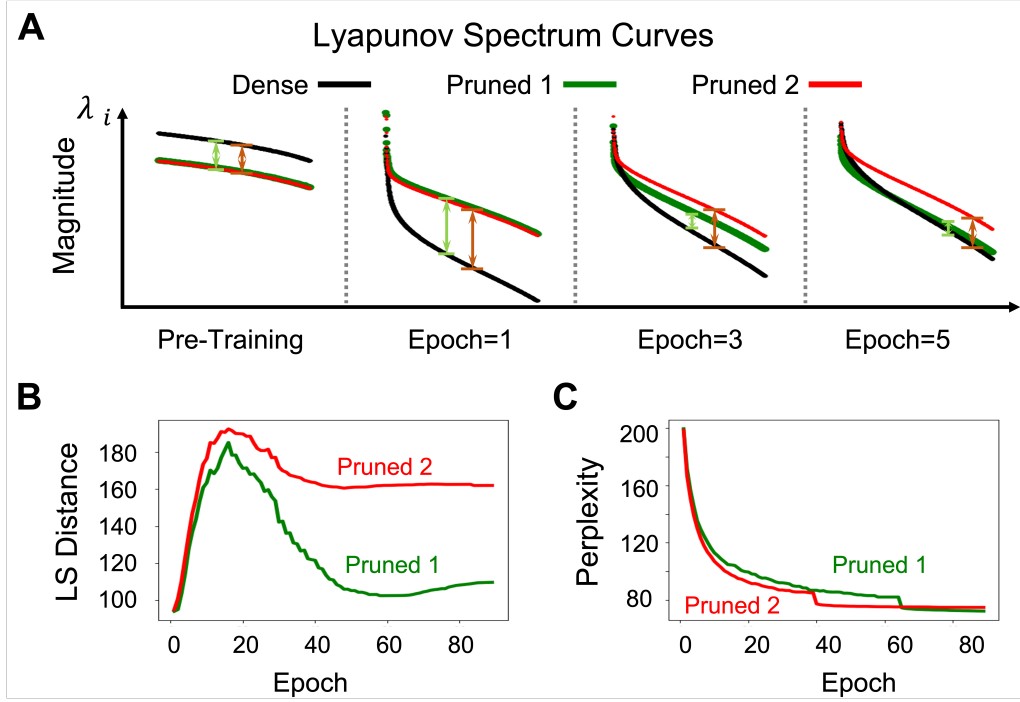

Figure 1: (A) Lyapunov Spectrum curves of dense (black), and two pruned variants (green and red) at pre-training, Epoch 1, 3, and 5 (left to right). (B) $L_2$ distance of two variants to the dense reference in LS space over training. (C) Perplexity curves for two pruned variants over training.

The Dynamic Sparse Training (DST) approach was introduced recently to meet the rising demand of optimizing computational costs for achieving pruned variants (Bellec et al., 2017). In contrast to dense-to-sparse approaches, DST is a *sparse-to-sparse* pruning method that starts with a sparse model and maintains the number of non-zero parameters over the entire training to improve not only the training speed along with the inference speed. DST involves three main procedural steps of pruning: weight removal, weight growth, and weight redistribution. For each step, salience (controls) such as magnitude- or gradient-based could be applied to decide how it is conducted. There are no universal controls applicable to all tasks and networks, as each has its intended scenario. This unique mapping between a scenario and a control in DST prevents generalizing a standard rule over all scenarios. Therefore, for a certain scenario, the controls which characterize the pruning method become additional key hyperparameters that need to be set such that pruning is executed in the most optimal way. We term this type of hyperparameter search as 'Hyperpruning'. Hyperpruning concerns with selecting both the pruning methodology with its controls along with other hyperparameters (related to training) for a particular scenario. Specifically, hyperpruning requires searching over methodological and non-methodological hyperparameters. Methodological hyperparameters define the pruning method, and non-methodological hyperparameters are independent of the pruning method and are applicable across them.

A unique feature of hyperpruning that does not typically hold for other hyperparameter optimization problems is the existence of estimated target accuracy since the optimal accuracy of its non-pruned (dense) counterpart is available and can serve as a loose upper bound for the pruned network to guide the hyperpruning process. However, even with this knowledge, searching through all pruning methods and their hyperparameters is a time-consuming task since each variant requires extensive training. Furthermore, there is no guarantee for achieving a more accurate configuration after investigating multiple unsuccessful configurations. Hyperparameter Optimization algorithms accelerate the search process by implementing a distance that could be optimal to either efficiently evaluate configuration variants or effectively generate reliable variants (Snoek et al., 2012; Hutter et al., 2011; Bergstra et al., 2011). In particular, such a distance will aim to provide *early estimation* of the accuracy of a considered configuration without proceeding with full training. This distance also targets to improve

the reliability of configurations by *adaptively generating* them according to the history of visited configurations and their corresponding distances. A commonly used distance for hyperparameter search is the loss curve from early training episodes, however, in many scenarios this distance is not effective since it is rarely guaranteed that models that perform well in the beginning of training will reach the desired accuracy at the end, as illustrated in Fig. 1-C. Therefore, finding an efficient and reliable distance is essential to hyperpruning.

Some previous work has shown that sequence models (RNNs and their variations) can be treated as dynamical systems, and their dynamic flow can be analyzed using Lyapunov Spectrum (LS), which measures the contraction/expansion ratio of the hidden states over time. It was also proposed that LS could be a potential indicator of the trainability of the network and the accuracy that it could reach (Vogt et al., 2022a). Based on these results, a recent work suggested that an AutoEncoder learned from the LS of slightly trained-RNN samples with random hyperparameter configurations can provide a low-dimensional embedding space in which RNN variants are organized according to the accuracy they achieve after training (Vogt et al., 2022b). Such organization appears to persist in the pruning setup establishing the embedding directly from LS, as demonstrated in Fig. 1-A. The example compares two pruned variants (green and red) with the dense model (black). Pruned 1 variant (green) eventually after full training achieves more optimal/lower perplexity than Pruned 2 variant (red) although its perplexity is higher for more than 60 epochs, see Fig. 1-C. While early perplexity is not indicative of the estimated accuracy, LS-based distance appears to be consistent and indicates that Pruned 1 variant is closer to the dense network for all epochs, as shown in Fig. 1-B.

The consistency of distance between the dense and pruned variants over training leads us to introduce a novel distance based on LS and its embedding (LS Space) to guide hyperpruning. Along with LS distance we propose a novel and efficient algorithm, called *LS*-based *H*yperpruning (**LSH**), which utilizes the distance in LS space (LS-based distance) and adaptive generation to identify optimal pruning variants at a fraction of the cost required for training them. We summarize the main contributions of our work below:

1. We propose a novel distance for hyperpruning. The distance is based on Lyapunov Spectrum (LS) and is capable of estimating the similarity between candidate pruned variants and the dense model.

2. We propose a novel algorithm called LSH that utilizes the LS-based distance as an early estimation criterion and allows sifting through and replenishing pruned variants with an order of magnitude smaller number of training epochs than loss-based full training search.

3. Our experiments on two extensive RNN language model benchmarks including stacked LSTM and RHN language models trained with Penn Treebank Dataset, show that LSH achieves more accurate pruned models than loss-based hyperparameter searches, pruning methods that achieved state-of-the-art accuracy on benchmarks, and even more accurate than the dense network.

## 2 RELATED WORK

**Network Pruning** The desire of reducing networks complexity and running time fuels the investigation of network pruning. In early applications, pruning was proposed to improve the inference speed of pretrained dense networks by iteratively pruning and fine-tuning them until reaching the target sparsity (Janowsky, 1989; Mozer & Smolensky, 1989; 1988; LeCun et al., 1989; Hassibi & Stork, 1992). Such pipelines are called *dense-to-sparse* and were applied in various applications and in many achieved only a marginal drop in accuracy when compared to the dense models, e.g. magnitude-based pruning applied to convolutional networks (Han et al., 2015b; Guo et al., 2016).

The requirement of a pre-trained model was mediated by Gradual Magnitude Pruning (GMP) methods which gradually train and prune the model over time and have been implemented on for broad range of network architectures (Narang et al., 2017; Zhu & Gupta, 2017; Gale et al., 2019). Recent approaches further proposed to generalize the pruning process such that the weights and the pruning mask are jointly learned through a unified optimization or differential reparameterization function (Liu et al., 2020; Kusupati et al., 2020; Lin et al., 2020). In addition to direct pruning on weights, indirect pruning methods have been proposed employing $L_0$ (Louizos et al., 2017), $L_1$ regularization (Wen et al., 2017), and Variational Dropout (Molchanov et al., 2017). However, for large-scale learning tasks these methods do not reach the accuracy of magnitude-based pruning (Gale et al., 2019).

Magnitude-based pruning turned out to be effective in several aspects. While pruned networks optimize the inference time and save computational resources the pruning process, as being a generalized iterative regularization, could also achieve more optimal accuracy than the dense model, as already observed in Lottery Ticket Hypothesis (Frankle & Carbin, 2018). In some cases the "lottery ticket" can be identified early in training and potentially save training cost (You et al., 2019). While GMP succeeds in achieving faster inference time, their training time is similar to or even longer than training a dense network. To improve the training efficiency and avoid complex pruning schedules, it has been proposed to prune at initialization according to particular salience (controls) and maintain the pruning mask throughout training (Lee et al., 2018; 2019; Wang et al., 2020; Tanaka et al., 2020). These methods are advantageous but fail to match the performance of dense-to-sparse methods (Wang et al., 2020) and typically do not perform well in extreme sparsity cases (Lee et al., 2019).

To address above challenges in expanding training demands, recent approaches proposed a novel type of pruning, Dynamic Sparse Training (DST). It is a *sparse-to-sparse* pruning concept which aims to optimize for both inference and training efficiency without sacrifice to accuracy. To keep being efficient, DST maintains the total number of non-zero weights during training and alters the position of non-zeros through a prune-regrow process to make the model more flexible. This idea was first introduced in Deep-R (Bellec et al., 2017), which rewires the network from a posterior. Later, inspired by biological neural networks, SET (Mocanu et al., 2018) simplified this process by replacing it with magnitude-based pruning and random weight growing to maintain the sparsity during training. Starting from a sparse seed network, NeST (Dai et al., 2019) prunes and grows both neurons and weights according to magnitudes and gradients, respectively. Consequently, DSR (Mostafa & Wang, 2019) introduced a non-uniform sparsity for different layers instead of uniform sparsity in SET. While showing promise, these sparse-to-sparse methods could not be on-par with dense-to-sparse methods. For further enhancement, SNFS (Dettmers & Zettlemoyer, 2019) proposed to use momentum for weight growing which achieves better accuracy than random growth and matches or even outperforms dense-to-sparse methods but at the price of computational efficiency (Dettmers & Zettlemoyer, 2019). To tackle the inefficiency, RigL (Evci et al., 2020) proposed to adapt "lazy gradient" calculation into magnitude-based pruning, and Top-KAST (Jayakumar et al., 2020) further optimized it by not requiring to calculate the dense gradients. Recently, a DST pruning method called Selfish-RNN has been proposed specifically for RNN. It showed a significant improvement over the performance of sparse RNN via non-uniform weight redistribution across gates and SNT-ASGD (Liu et al., 2021). The method addresses parameter allocation among different layer types, which was also observed as key in other works (Kusupati et al., 2020; Frankle et al., 2020). Additional comprehensive survey on network pruning is available in (Hoefler et al., 2021; Wang et al., 2021) and references therein.

***Hyperparameter Search*** Multiple Hyperparameter Optimization (HPO) algorithms have been proposed to find the optimal hyperparameter configurations for a specific scenario. Many HPO algorithms propose configurations in an iterative fashion and then evaluate their performance. Therefore, HPO can be categorized the way they *propose* new configurations and how they *evaluate* configurations.

For configurations proposal, grid search and manual search are the most intuitive HPO, however are also inefficient. Instead, random search finds comparable or better models within a limited time (Bergstra & Bengio, 2012). In contrast to these memory-less approaches, more advanced HPO algorithms include a memory of proposed configurations and their accuracy in an archive. They then decide between exploitation and exploration, i.e., exploiting the current best configuration (local search) or exploring a new random configuration (random search). Bayesian Optimization, such as TPE, uses the archive to fit a surrogate model and then adaptively finds configurations (Snoek et al., 2012; Hutter et al., 2011; Bergstra et al., 2011). Given this extra information from the archive, Bayesian Optimization empirically outperforms a plain random search (Thornton et al., 2013; Eggensperger et al., 2013; Snoek et al., 2015).

Another variation of HPO is multifidelity search which focuses on accelerating configuration evaluation, i.e., fast evaluation is used to infer the configuration performance (Bischl et al., 2021). This approach aims to find promising candidates and allocate more resources to them. Typically, there is a trade-off between computational cost and evaluation accuracy, i.e., the more computation spent on evaluating a configuration, the more accurate the evaluation is. Therefore, one common practice is to remove less promising candidates. This approach can be incorporated into random search or other HPO to explore a more expansive searching space within the same resource budget (Li et al., 2017). Searching over a larger space is necessary for complex architectures, such as sparse stacked LSTM and RHN, due to many possible hyperparameter configurations. Therefore, a fast and effective

configuration evaluation method becomes essential. Here we show that Lyapunov Spectrum (LS) is a candidate for such method since able to effectively perform early estimation in training.

***Lyapunov Spectrum*** Treating sequence models, such as RNN as dynamical systems and predicting their long-term performance is an emerging topic in deep learning research (Chang et al., 2019; Zheng & Shlizerman, 2020; Vogt et al., 2022a; Ribeiro et al., 2020). Lyapunov Spectrum (LS), formed by Lyapunov Exponents, is targeted to identify nonlinear dynamical systems characteristics (Ruelle, 1979; Oseledets, 2008). LS was initially used to provide insight into autonomous neural network models (Monteforte & Wolf, 2010; Engelken et al., 2020), while RNNs are typically non-autonomous due to external inputs. However, as long as input sequences are sampled from a stationary distribution, LS can be generalized to a non-autonomous dynamical system such as RNN based on random dynamical theory (Arnold, 1995). Another property of LS is that if RNN sequence is long enough, the resulting LS converges to the same curve, as proved by Oseledets theorem (Saitô & Ichimura, 1979; Ochs, 1999). These results indicate that the information captured by LS could be indicative of network dynamics. In (Legenstein & Maass, 2007; Pennington et al., 2018; Laurent & von Brecht, 2016), the correlation between the largest Lyapunov exponent sign and the existence of chaos in the network has been studied. In addition, research succeeded in connecting LS features, such as zeros, negative, mean, and variance, and concepts in dynamical systems, such as quasi-periodic orbits, fixed point attractor, heterogeneity, and rate of contraction (Dawson et al., 1994; Abarbanel et al., 1991; Shibata, 2001; Brandstäter et al., 1983; Yamada & Ohkitani, 1988). While each feature seems to extract knowledge about the network, their correlation with network accuracy is unclear (Vogt et al., 2022a). With the efficient LS computing algorithm proposed in (Vogt et al., 2022a), the correlation between LS and network performance was found via an additional AutoEncoder network (Vogt et al., 2022b) such that this AutoEncoder can be used to predict the performance of random hyperparameter configurations. Motivated by this work, we propose an LS-based distance that is used to eliminate unpromising candidates and generate new candidates. While in general applications an additional AutoEncoder network might be needed for organizing candidates into similarity groups, for pruning, the existence of a dense model for which LS can be computed sets the similarity and we find that direct distance between LS of dense and pruned networks provides a robust indicator of their similarity.

## 3 METHODS

LS-based Hyperpruning (LSH) both eliminates and generates candidates. LSH effectively eliminates unpromising candidates from the candidate set based on their closeness to the dense reference in the LS space. It then generates new candidates given the remaining candidates in the set and inserts them into the set. Such elimination and insertion procedure is iterated for multiple times. At completion, the most promising candidates are selected with most of the training resources allocated to them for full extensive training. The LSH method is illustrated in Fig. 2.

***Lyapunov Spectrum Computation*** We adapt the algorithm from (Vogt et al., 2022b), which computes LS by monitoring the contraction/expansion history of the network over the entire time sequence. Specifically, it first randomly selects a batch $\mathcal{X}$ of $K$ samples with sequence length $T$ from the validation set $\mathcal{D}_{val}$, i.e., $\mathcal{X} \equiv \{\mathbf{x}^k = \{x^{t,k}\}_{t=1}^T, \mathbf{x}^k \in \mathcal{D}_{val}, k = 1, 2, \ldots, K\}$, where $x^{t,k}$ is the $t$-th time step of the $k$-th sample in $\mathcal{X}$. A vector of zeros $\mathbf{0}$ and an identity matrix $\mathcal{I}$ are initialized as the original hidden state $h^0$ and orthogonal representation base $\mathcal{B}^0$. Hidden state $h^t$ tracks the evolution of network and orthogonal representation base $\mathcal{B}^t$ captures contraction/expansion of each direction in the hidden state till time step $t$. At each time step, $h^t$ and $\mathcal{B}^t$ are updated, i.e., $h^{t,k} = f(h^{t-1,k}, x^{t,k})$ and $\mathcal{B}^{t,k}, R^{t,k} = qr(J^{t,k} \cdot \mathcal{B}^{t-1,k})$, where $f$ is the network function, $J$ is the Jacobian matrix between hidden states, $qr$ denotes QR decomposition, and $\cdot$ denotes matrix multiplication. The contraction/expansion ratio at time step $t$ is incorporated into the $i$-th vector of $\mathcal{B}^t$ and measured by $r_i^t$, the $i$-th diagonal element of $\mathcal{R}^t$. The $i$-th $LS$, $\lambda_i$ is then calculated by taking the average over the entire time sequence $T$ and $K$ samples.

$$\lambda_i = \frac{1}{K}\frac{1}{T}\sum_{k=1}^{K}\sum_{t=1}^{T} log(r_i^{t,k}) \tag{1}$$

We denote LS as $\Lambda$, i.e., $\Lambda \equiv \{\lambda_i\}_{i=1}^L$, where $L$ is the dimension of hidden states $h^t$.

***LS-based Hyperpruning (LSH)*** Starting with a random candidate set $\mathcal{T}$ where each candidate is a hyperparameter configuration of a pruned network, the goal of hyperpruning is to find the best

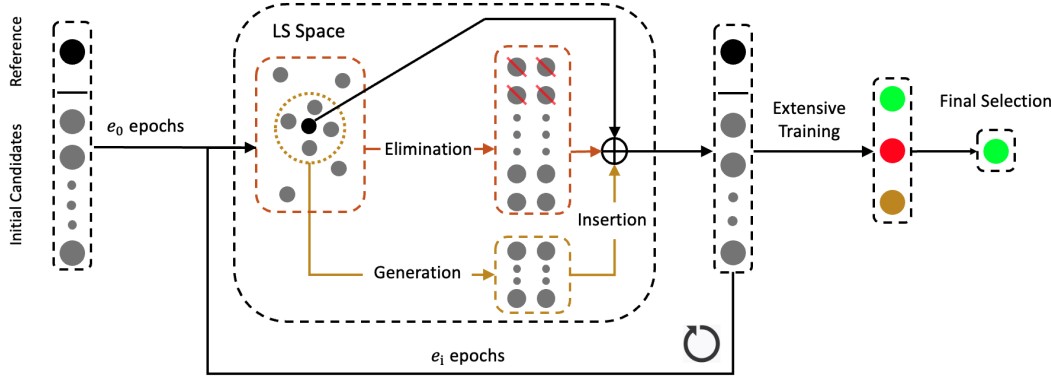

Figure 2: LS-based Hyperpruning: The process starts with an initial candidate set of hyperparameter configurations (gray circles) and a dense reference model (black circle), and executes elimination and insertion in turns. Elimination: The reference model and each candidate is trained for $e_0$ or $e_i$ epochs and projected to the LS space. Candidates are ranked and eliminated depending on their distance to the reference in the LS space. Insertion: New candidates are adaptively generated given the remaining candidate in the set and inserted into the set. After several rounds of elimination and insertion, the remaining candidates in the set are extensively trained and the best one (green circle) is picked as the final selection.

configuration, see Fig. 2. The procedure of LSH is guided by a dense network $\hat{t}$ since it provides a target performance for the pruned model to reach or even surpass it. Candidate selection process includes LS-based elimination and adaptive insertion phases executed in turns for several iterations. In the LS-based elimination step, each candidate $t^i \in \mathcal{T}$ is trained for $E$ epochs, and after each epoch of training, their LS is computed according to Eq. 1 and denoted as $\Lambda_j^i$ where $i$ and $j$ stand for the $i$-th candidate and the $j$-th epoch, respectively. LS till the $E$-th epoch of $t^i$ are grouped as $\Lambda^i$, i.e., $\Lambda^i \equiv \{\Lambda_j^i\}_{j=0}^E$. Similarly, all LS from the dense model till epoch $E$ are denoted as $\hat{\Lambda}$. We then project $\hat{\Lambda}$ and $\Lambda^i$ to an embedding space and use the distance between the last epoch of $\hat{t}$ and $t^i$ to indicate their closeness, i.e., $[\hat{v}_0, \ldots, \hat{v}_E, x_0^i, \ldots, v_E^i] = embedding([\hat{\Lambda}, \Lambda^i])$ and $s = distance(\hat{v}_E, v_E^i)$, where $E$ can either be the starting epoch $e_0$ or the incremental epoch $e_i$ depending at which time the LS-based elimination is being performed, see Fig. 2. In our work, we choose PCA to construct the embedding space and project LS to a 2d PCA space, called LS Space. We use $L_2$ distance to measure the closeness of networks in this LS space. The distance $s$ between $t^i$ and $\hat{t}$ is then used to rank the candidates in $\mathcal{T}$. In our case, $\frac{n}{2}$ out of $n$ candidates with longer distances are eliminated from $\mathcal{T}$. Even though LS-based elimination can efficiently find the best configuration in $\mathcal{T}$, it is not guaranteed to find the best configuration for this task since the best configuration may not necessarily exist in the initial set. Therefore, in addition to elimination, LSH adaptively generates new candidates based on the remaining candidates in $\mathcal{T}$ and inserts them to $\mathcal{T}$. Generation is based on remaining promising candidates since as LSH is iterating through elimination, it refines the candidates such that promising candidate would move the distribution in $\mathcal{T}$ closer to the optimal distribution and would increase the chance of finding the best configuration. The new candidate generation and insertion happens after the LS-based elimination. Specifically, after the elimination, given the remaining $\frac{n}{2}$ candidates in $\mathcal{T}$, the knowledge of their configurations and distances $s$ serves as an initial history/archive for Bayesian hyperparameter optimization methods such as Tree Parzen Estimators (TPE) and Adaptive TPE (APTE) to generate new candidates. For example, TPE uses this initial knowledge to build a surrogate model and generate new configuration candidates via Expected Improvement. In particular, $\frac{n}{4}$ new candidates are generated and then added to candidate set $\mathcal{T}$. This uneven number of elimination($\frac{n}{2}$) and insertion($\frac{n}{4}$) candidates from and to $\mathcal{T}$ asymptotically decreases the number of candidates in $\mathcal{T}$ so that eventually only the most promising candidates are left in $\mathcal{T}$. The elimination/insertion schedule is controlled by $e_0$ and $e_i$ which decide when to execute the first and future elimination, respectively. As a result, they manipulate the required time budget for the candidate selection process.

After $m$ epochs of the candidate selection process, the remaining candidates in $\mathcal{T}$ are extensively trained for $P$ epochs where $P >> m$ until their losses/accuracies converge. Given the final performance of extensively trained candidates, the best one is selected as the final pruned network.

## 4 EXPERIMENTS

To demonstrate the effectiveness of *LS-based* distance (LSH), we perform multiple experiments comparing *LS-based* distance (LSH) and *Loss-based* distance as early estimation criteria for HPO algorithms in hyperpruning. The same initial candidates set size and elimination/insertion schedule are used for a fair comparison, i.e., the number of initial candidates $n$, the starting epoch $e_0$, and incremental epoch $e_i$ are kept the same. This results in the same resource budget for both methods since it only depends on $n$, $e_0$, and $e_i$. We show the generality of our method by applying it to two sequence models, stacked LSTM and RHN, trained with Penn Treebank dataset (Marcinkiewicz, 1994) for a language modeling task. In each experiment, we follow the setup of a SOTA RNN pruning (Liu et al., 2021). We also show the robustness of LSH by conducting experiments on stacked LSTM under different pruning ratios. Furthermore, we demonstrate the time efficiency of LSH by comparing the time difference between LSH and the loss-based full training search to achieve a target accuracy. The training and LS computation are conducted on an NVIDIA RTX 2080 Ti. More hyperparameters of architectures can be found in Supplementary Material.

Table 1: The validation and testing perplexity of stacked LSTM and RHN on PTB dataset for a language modeling task (lower perplexity means better performance). LSH applied with Grid Search (GS), TPE, and Adaptive TPE (ATPE). Vanilla GS, TPE, and ATPE use Loss-based distance. DSR, SNFS, SET, RigL, GMP, Dense, and Selfish-RNN data are from (Liu et al., 2021).

| Method | Stacked-LSTM | | RHN | |
|---|---|---|---|---|
| | Validation | Testing | Validation | Testing |
| DSR (Mostafa & Wang, 2019) | 90.0 | 88.2 | 65.4 | 63.2 |
| SNFS (Dettmers & Zettlemoyer, 2019) | 88.3 | 86.3 | 74.0 | 71.0 |
| SET (Mocanu et al., 2018) | 87.3 | 85.8 | 63.7 | 61.1 |
| RigL (Evci et al., 2020) | 78.3 | 75.9 | 64.8 | 62.5 |
| GMP (Gale et al., 2019; Zhu & Gupta, 2017) | 76.8 | 74.8 | 65.6 | 64.0 |
| Dense (Liu et al., 2021) | 74.5 | 72.4 | 63.4 | 61.8 |
| GS | 74.2 | 72.5 | 62.1 | 60.4 |
| APTE | 74.3 | 72.7 | 63.4 | 61.6 |
| TPE (Bergstra et al., 2011) | 74.4 | 72.7 | 62.1 | 60.2 |
| Selfish RNN (Liu et al., 2021) | 73.8 | 71.7 | 62.1 | 60.4 |
| LSH-GS (Ours) | 72.5 | 70.8 | 62.1 | 60.4 |
| **LSH-ATPE (Ours)** | **72.1** | **69.9** | **60.5** | **59.5** |
| **LSH-TPE (Ours)** | **72.0** | **69.9** | **60.2** | **59.0** |

***Effectiveness.*** We first demonstrate the effectiveness of LSH for selecting hyperparameter configurations including methodological and non-methodological hyperparameters. We consider sparse initialization, death mode, and redistribution mode for the methodological hyperparameters. Sparse initialization determines the initial pruning mask. Death and redistribution modes determine how the weights in the network are pruned and redistributed before and after weight growth, respectively. There are two available initialization modes, four death modes, and three redistribution modes. For the non-methodological hyperparameters, we consider the death rate, which is the ratio of pruned weight and total nonzeros weight at each time. The death rate is generated from the range of $[0.4, 0.9]$.

We evaluate LS-based and Loss-based distances on different HPO algorithms, such as Grid Search (GS), TPE, and Adaptive TPE (APTE). The pruning ratio for Stacked LSTM and RHN is $0.67$ and $0.53$, respectively. $e_0$ and $e_i$ are $[3, 3]$ and $[6, 3]$ for stacked-LSTM and RHN, respectively. For GS, we fix the death rate of stacked LSTM to $0.8$ and RHN to $0.5$. Therefore, only methodological hyperparameters are considered and result in a total of $24$ configurations. For TPE and APTE in Table. 1, $n$ is $40$ for both stacked LSTM and RHN. As shown in Table 1, LSH (bottom three rows) consistently outperforms by a margin dense models and SOTA pruning methods. Note that in Table. 1 is that GS, TPE, and APTE row uses Loss-based distance in comparison with LS-based distance (bottom three rows). LSH with different HPO algorithms also achieves better performance than the loss-based counterparts in most cases (3/3 in stacked LSTM and 2/3 in RHN). The best result is achieved by LSH-TPE in both stacked LSTM and RNH cases which outperforms the previous best result (Selfish RNN), i.e., 69.9 vs. 71.7 for stacked LSTM and 59.0 vs. 60.4 for RHN. Another

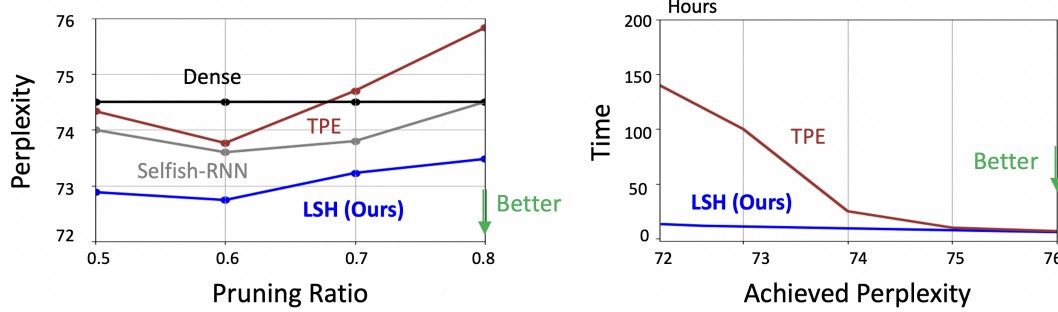

Figure 3: (Left) Pruning Ratio vs. Perplexity: Comparison among LSH (blue), TPE/loss-based (brown), and selfish-RNN (gray) on different pruning ratios.; (Right) Achieved Perplexity vs Time Budget: Comparison between LSH (blue) and loss-based full training search (brown) on required time budget (hours) to reach a target perplexity.

important observation is that LSH with HPO algorithms, such as TPE and ATPE, outperforms LSH with GS. Quantitatively, LSH obtains 0.9 and 1.4 perplexity improvement on stacked LSTM and RHN by switching from GS to TPE, respectively. Notably, this improvement is not found in the loss-based counterparts and indicates that LSH utilizes advanced hyperparameter optimization methods better for forming more optimal elimination and generation which leads to a more optimal selection.

We also test LSH under different resource budgets, i.e, different sizes of the initial candidate set. We consider three resource budgets, low, moderate, and high which have 24, 30, and 40 candidates in the initial set, respectively. We demonstrate it on stacked LSTM with pruning ratio= 0.67. Each experiment is repeated three times and the average perplexity with 95% confidence interval are reported. As shown in Table. 2, with increasing resource budget (Low → High), LSH can improve the average performance (70.9 → 69.9 for TPE and ATPE) and achieve more reliable result as confidence interval drops (3.0 → 2.6 for TPE and 0.4 → 0.3 for ATPE), while this improvement does not necessarily exist in Loss-based counterpart (mean: 72.3 → 72.7 for TPE 72.0 → 72.7 and ATPE; confidence interval: 0.8 → 1.1 for TPE and 1.9 → 0.1 for ATPE).

Table 2: Comparison between LS-based (LSH) and Loss-based distances applied with GS, TPE, and ATPE under different budgets. Stacked LSTM with pruning ratio 0.67 trained on PTB is used here. The average perplexity with 95% confidence interval are reported.

| Methods | Low Budget | | Moderate Budget | | High Budget | |
|---|---|---|---|---|---|---|
| | Validation | Testing | Validation | Testing | Validation | Testing |
| GS | 74.2 | 72.5 | 74.2 | 72.5 | 74.2 | 72.5 |
| ATPE | 74.2 (1.2) | 72.0 (1.9) | 74.3 (0.7) | 72.7 (0.1) | 74.3 (0.7) | 72.7 (0.1) |
| TPE | 74.1 (1.3) | 72.3 (0.8) | 74.2 (0.6) | 72.4 (1.0) | 74.4 (0.5) | 72.7 (1.1) |
| **LSH-ATPE (Ours)** | **73.3 (0.9)** | **70.9 (0.4)** | **72.1 (0.2)** | **69.9 (0.3)** | **72.1 (0.2)** | **69.9 (0.3)** |
| **LSH-TPE (Ours)** | **72.6 (1.9)** | **70.9 (3.0)** | **72.8 (1.1)** | **70.9 (2.6)** | **72.0 (0.2)** | **69.9 (2.6)** |
| **LSH-GS (Ours)** | **72.5** | **70.8** | **72.5** | **70.8** | **72.5** | **70.8** |

***Robustness.*** We evaluate the robustness of LSH to pruning ratios and show our results in Fig. 3-Left. We test from moderate (0.5) to high (0.8) pruning ratios by a step of 0.1. For each experiment, TPE is used as the HPO algorithm and $n$ is set to 20. The experiment is on Stacked LSTM and the same $e_0$ and $e_i$ are used. As shown in Fig. 3-Left, LSH (blue) consistently outperforms TPE/loss-based (brown), Selfish-RNN (gray) methods, and dense model for all pruning ratios.

***Time Efficiency.*** We compare the time that LSH uses to reach a particular target perplexity with the baseline loss-based full training search, which selects the optimal configuration after finishing training all candidates. We use stacked LSTM with a fixed pruning ratio of 0.67 for these experiments. Fig. 3-Right shows the amount of time (y-axis) required for finding a configuration with a particular target perplexity (x-axis). We observe that LSH is more efficient than the loss-based approach for all perplexities. Particularly, for finding the optimal configuration (72), LSH (≈15 hours) is an order of magnitude faster than the loss-based approach (≈150 hours, > 6 days).

***Ablation Study.*** For the completeness of LSH, we perform ablation studies of the effect of elimination/insertion schedule and LS computation efficiency. Experiments are on stacked LSTM.

We first studied the efficiency and effectiveness of elimination/insertion schedules set by $e_0$ and $e_i$. We fixed the hyperparameter optimization method to Grid Search with 24 configurations, and the rest of the setup is as in previous experiments. Table. 3 shows the number of epochs required for the candidate selection process under each schedule. Cells with green color denote that the optimal configuration was indeed selected. We find that with $e_0 = 3, e_i = 1$, the minimum number of required epochs is achieved (130), the performance might not be robust across different experiments since, with a small change in the $e_0$ direction, the selection might fail. Therefore, we set $e_0 = e_i = 3$ which is robust is both $e_0$ and $e_i$ directions. While this configuration requires addition 116 epochs (an increase from $130 \rightarrow 246$) which is about 2 hours of additional running time, this time is still significantly shorter compared to extensive training of the two remaining candidates ($> 20$ hours).

Table 3: The efficiency and the effectiveness of scheduling. Each cell reports the number of required epochs for the candidate selection process. Green cells mean the best candidate is indeed selected.

| $e_0$ \ $e_i$ | 1 | 2 | 3 | 4 |
|---|---|---|---|---|
| 1 | 82 | 140 | 198 | 256 |
| 2 | 106 | 164 | 222 | 280 |
| 3 | 130 | 188 | 246 ($\star$) | 304 |

Notably, LS computation is an efficient process and is relatively faster than training the network itself, i.e., it takes 6s for the LS computation of each sample vs. 120s for one training epoch. However, with large LS computation batch sizes, the overall process could slow down hyperpruning. Here we empirically show that only a few validation samples are needed for the LS computation. We demonstrate it by comparing the maximum, mean, and variance of LS computed from different numbers of validation samples across two pruned models. As discussed in (Vogt et al., 2022a), those are the main features that determine the characteristic of LS. As shown in Table. 4, maximum, mean and variance are similar for LS calculated from 2 and 10 validation samples. Their different $\Delta$ is one order of magnitude smaller compared to the difference between different pruned models.

Table 4: Effect of LS computation batch size: The maximum, mean, and variance of LS computed from 2 and 10 validation samples on two pruned models.

| | Pruned 1 | | | Pruned 2 | | |
|---|---|---|---|---|---|---|
| | 2 | 10 | $\Delta$ | 2 | 10 | $\Delta$ |
| maximum | -1.26 | -1.20 | 0.06 | -0.52 | -0.46 | 0.06 |
| mean | -4.40 | -4.41 | 0.01 | -3.97 | -3.97 | 0 |
| variance | 2.59 | 2.67 | 0.08 | 1.92 | 1.98 | 0.06 |

# 5 CONCLUSION

In this work, we propose a novel hyperparameter search approach that deals with selecting the optimal pruning method with the corresponding hyperparameter configuration for a particular scenario. To resolve that, we propose a novel Lyapunov Spectrum(LS)-based hyperpruning algorithm, termed LSH, to effectively and efficiently select the optimal configuration. LSH iteratively eliminates unpromising pruned variants based on their distance to a dense reference model in LS space and adaptively generates new variants that are enhanced iteratively. Eventually, most resources are allocated to the most promising selected candidates to fully train an optimal pruned network. Our method can be incorporated into different existing hyperparameter optimization algorithms, such as TPE and ATPE, and achieves an order of magnitude boost compared to the loss-based full training search in terms of efficiency. We conduct experiments on stacked LSTM and RHN models trained with the Penn Treebank Dataset on language modeling tasks. Our results show that LSH is robust to different pruning ratios and given a fixed pruning ratio, it outperforms its loss-based counterparts, other SOTA pruning method, and even the dense model.

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
