# OpenReview forum: "HYPERPRUNING: EFFICIENT PRUNING THROUGH LYAPUNOV METRIC HYPERSEARCH"
_ICLR.cc/2023/Conference — Submitted to ICLR 2023_

### Official Review · Reviewer_X75X · 2022-10-24

**Confidence:** 3
**Correctness:** 3
**Technical Novelty And Significance:** 2
**Empirical Novelty And Significance:** 1
**Recommendation:** 3

**Clarity, Quality, Novelty And Reproducibility:**

Clarity:
The clarity of this paper can be improved: see weakness 3.

Quality:
The discussion of main technical components are missing: see weakness 1 and 2.

Novelty:
I am not expert of neither RNN pruning or LS literature, but I think this paper is novel in that it is the first one which introduces LS-distance for RNN pruning.

Reproducibility:
This paper provides all the experimental setups to reproduce results.




**Strength And Weaknesses:**

Strength:
1. The main idea of this paper is intuitive: using LS-distance as a metric for eliminating unpromising sparse network candidates. It naturally supports the usage of reference (dense) model as a target, which is ambiguous for the original Loss-based metric.
2. The experimental results show the effectiveness of the proposed method.

Weakness:
1. The computational overhead of Lyapunov Spectrum is not discussed in the paper. In equation 2, QR decomposition and Jacobian matrix computation are required. This makes the proposed method more computation- and memory-intensive than Loss-based metric.
2. The reason why one should use 2D-PCA as a embedding function is rarely discussed. For instance, one can use either tsne embedding instead of 2D-PCA or the LS-distance itself without any modifications. At least, the reason why authors set 2D for PCA can be discussed.
3. The writing can be improved, e.g., some paragraphs are too long, notations are overlapped (x is used for both input and embedding).
4. The experiment is too narrow to validate the general effectiveness of proposed method. RNNs are widely used for tons of NLP tasks (e.g. machine translation as explained in introduction), and the proposed method is not limited to any specific NLP tasks.
5. I am not expert of neither RNN pruning or LS-distance, but I think this method can be applied to Transformer-based neural networks. Transformer is recently widely used for any NLP tasks, and pruning is one of attractive research directions for these large-scale models.

**Summary Of The Paper:**

- The motivation of this paper is that Lyapunov Spectrum (LS)-based distance in the early training stage can represent the final performance.
- Based on the above observation, this paper illustrates hyperparameter optimization method for RNN pruning using LS distance.
- Specifically, the candidate set of configurations (i.e., sparse RNN) are initialized and trained for E epochs.
- The LS is computed for each candidate and dense model through training: eq 1, 2, and 3.
- All the LS values of candidates and dense models over training epochs are then used to obtain embedding (e.g. PCA).
- Then the candidate is eliminated based on the L2 distance between candidate embedding and dense model embedding of the last epoch.
- The experiments are conducted on PTB dataset using stacked-LSTM and RHN.
- The proposed method outperforms baselines including dense model.

**Summary Of The Review:**

This paper proposes a RNN pruning method, which eliminates sparse network candidates based on LS-distance metric. The idea is novel and motivating, but the discussion of main technical components are missing and experiments are too narrow to evaluate of general effectiveness of the proposed method.

---

> ### Author Response · Authors · 2022-11-16
> **Response to the Reviewer X75X**
>
> **Response**
>
> We thank the reviewer for a thoughtful review and valuable feedback. We provide point-by-point clarifications and answer questions raised by the reviewer below.
>
> **Strength and Weaknesses:**
>
> **W1. Computational Overhead of LS:** We thank the reviewer for pointing out the need for further discussion of Lyapunov Spectrum (LS) computation. It is correct that LS computation could be computationally expensive and memory heavy since it involves QR decomposition and Jacobian matrix multiplication in comparison to computing the perplexity for example. In our experiments, we use an NVIDIA GeFORCE RTX 2080 Ti GPU and it takes about 6s to compute the LS computation for each sample. However, as we mention in Table. 4 of the manuscript, the difference between using 2 and 10 samples is not significant, so in our experiments, we just use two samples for the LS computation which takes about 12s. Compared with the time of training an epoch (90s), LS computation is only a fraction of this time and slows down the candidate selection process by 13% compared to the perplexity-based one. Note that the extensive training process does not require LS computation. Given the effectiveness of LSH in choosing more accurate candidates, this efficiency tradeoff is reasonable since saves hours of extensive training as we show in Figure 3 (Right) in the manuscript. Moreover, for larger problems, even if the LS computation is expensive since only a few samples are required, the computation overhead resulted by the LS computation is still acceptable compared to the whole training. We include this additional discussion in Supplementary Material section D.
>
> | Embedding | PCA | PCA | LS | LS  | T-SNE |  T-SNE|
> |:---:|:---:|:---:|:---:|:---:|:---:|:---:|
> | Distance Metric | L2 | Cos | L2 | Cos | L2 | Cos |
> | Perpelxity | **69.9** | 72.99 | 72.56 | 70.3 | 70.5 | 72.6 |
>
> **W3. Manuscript Clarity:** We thank the reviewer’s feedback on the length of some paragraphs and notations usage. In the updated manuscript, we modified the structure of the paper to make it easier to follow. Specifically, we now divide the EXPERIMENTS section into two parts: (i) description of the experimental setup (ii) description of the results. In terms of notations, we will use x for inputs and v for embedding to avoid conflict.
>
> **W4. Task Generalization:** The experiments were conducted on a language modeling task which is used as a standard task to evaluate the model's ability to extract useful information from a wide–range context. Multiple works have addressed this problem and task, e.g. [2, 3, 4]. Furthermore, previous RNN pruning work [5] focuses on this task as well. Notably, there are no special constraints for LSH to this language modeling task and therefore the approach is directly generalizable to other tasks (NLP and other domains).
>
> **W5. Generalization to Transformer Architecture:** We thank the reviewer for suggesting performing LSH on SOTA NLP networks, such as Transformer-based neural networks. Indeed, one of the prospective directions would be to apply LSH to Transformers which are widely used SOTA language models. This is beyond the scope of this work since in this work we wanted to establish that LSH is able to achieve better accuracy than other pruning methods by hyperparameter search. The experiments we did on Stacked LSTM and RHN trained with Penn Tree Bank dataset are commonly used as benchmarks in RNN pruning literature [3]. Notably, there are no special constraints on our approach to those two architectures or PTB dataset and therefore this approach is expected to be generalizable to other models or datasets.
>
>
> [1] Ryan Vogt, Yang Zheng, and Eli Shlizerman. Lyapunov-guided embedding for hyperparameter selection in recurrent neural networks. arXiv preprint arXiv:2204.04876, 2022b.
> [2] Wojciech Zaremba, Ilya Sutskever, and Oriol Vinyals. Recurrent neural network regularization.arXiv preprint arXiv:1409.2329, 2014.
> [3] Pankaj Malhotra, Lovekesh Vig, Gautam Shroff, Puneet Agarwal, et al. Long short term memory networks for anomaly detection in time series. In Proceedings, volume 89, pp. 89–94, 2015
> [4] Julian Georg Zilly, Rupesh Kumar Srivastava, Jan Koutnık, and J̈urgen Schmidhuber. Recurrent highway networks. In International conference on machine learning, pp. 4189–4198. PMLR, 2017.
> [5] Shiwei Liu, Decebal Constantin Mocanu, Yulong Pei, and Mykola Pechenizkiy. Selfish sparse rnn training. In International Conference on Machine Learning, pp. 6893–6904. PMLR, 2021.

---

### Official Review · Reviewer_cX2r · 2022-10-26

**Confidence:** 2
**Correctness:** 3
**Technical Novelty And Significance:** 3
**Empirical Novelty And Significance:** 3
**Recommendation:** 6

**Clarity, Quality, Novelty And Reproducibility:**

The paper is overall clear, but the reference format doesn’t align with the ICLR requirement. The authors can double-check how to make the reference format correct. Moreover, some references are missing, e.g., in the second paragraph of the introduction, when introducing Dynamic Sparse Training (DST), authors need to cite the relevant papers.

The quality is moderate. I’m not very familiar with Lyapunov Spectrum (LS) or the language tasks. So it is hard to evaluate the novelty.


**Strength And Weaknesses:**

Using Lyapunov Spectrum for hyperparameter learning is an emerging method in the deep learning area. In this paper, the author proposed to use the Lyapunov Spectrum for channel pruning. The results show the proposed method achieves higher results than the SOTA.

I’m curious how is the LS distance compared with the L2 distance? What is the performance difference compared to using L2 distance?


**Summary Of The Paper:**

This paper proposed a new distance based on Lyapunov Spectrum (LS) for neural network hyperpruning. Here, the authors define ‘hyperpruning’ as finding a suitable pruning method with optimal hyperparameter configuration. The authors apply this method to RNN language model benchmarks and show good results on Penn Treebank Dataset.

**Summary Of The Review:**

Overall this paper proposed a plausible method that achieves good results on language tasks.

---

> ### Author Response · Authors · 2022-11-16
> **Response to the Reviewer cX2r**
>
> **Response**
>
> We thank the reviewer for a thoughtful review and valuable feedback. We have addressed the comments raised by the reviewer and revised the manuscript accordingly. We describe the revisions and provide point-by-point clarifications below.
>
> **Strength and Weaknesses:**
>
> **W1. L2 Distance:** We thank the reviewer for suggesting L2 instead of LS distance. To address this question and test which distance could be more effective, we have conducted an ablation study on different distances (L2, LS, Cos). Table. 1 below shows the perplexities of the final configurations selected by L2 and Cosine distance metrics. Specifically, for L2 and Cosine distance, we use pruned model LS and dense model at the latest epoch E, and calculate the distance between them. As we can see, LS-based distance achieves a better result (lower perplexity) than L2 and Cosine distances. We include this ablation study in Supplementary Material section E.
>
> Table 1: Perplexities of the final configuration selected based on different distance metrics.
> | Distance | LS | L2 | Cosine |
> |:---:|:---:|:---:|:---:|
> | Perplexity | **69.9** | 70.5 | 72.6 |
>
> **Clarity, Quality, Novelty and Reproducibility**
>
> **C1. Misaligned Reference Format:** We thank the review for noticing the incompatibility between our reference format and ICLR format. It turns out to be a typo in using a different LaTex \bibliographystyle. We have changed the style to ICLR23 and now it aligns with the ICRL reference format. We made this change in the updated manuscript.
>
> **C2. Missing References:** Thank you for pointing out that references have not been cited at their first occurrence in the introduction. We fixed it and cited these corresponding references in the right place in the revised manuscript.

---

### Official Review · Reviewer_mZ4i · 2022-10-27

**Confidence:** 4
**Correctness:** 3
**Technical Novelty And Significance:** 3
**Empirical Novelty And Significance:** Not applicable
**Recommendation:** 3

**Clarity, Quality, Novelty And Reproducibility:**

* Clarity:
The introduction is very clear, the section on LS calculation and LS-based hyperpruning could use more details. For example, it doesn't become clear to me, how exactly new candidates are being generated. Which features exactly are being fed into TPE or ATPE?

Quality:
The submission contains a good amount of work on comparisons with previous methods. (However, I am not an expert on efficient pruning, so I can't fully certify the quality of that part.
The Quality of the LS-based hyperpruning is hard to estimate, because no code is provided, so it is difficult to evaluate how the embedding and LS calculation were done exactly.
* For table 1, no confidence intervals are reported.

Novelty:
The proposed hyperpruning method seems novel. It seems an innovative addition to previous methods (e.g., Selfish RNN, APTE, GS) that seems to improve performance and reduce computational demand.

Reproducibility:
Because details are missing, and no code is provided, the results are unfortunately not reproducible in the current form.


**Strength And Weaknesses:**

Strength:
* The proposed hyperpruning method is novel.
* The submission bridges tools from different fields (network pruning, Lyapunov Spectrum, Language modelling)

Weakness:
1.) Many details are missing, which makes the submission hard to evaluate and impossible to reproduce. E.g.
- How exactly are the variants generated? The relevant paragraph says "after the elimination, given the remaining n candidates in T, the knowledge of their configurations and distances s serves as an initial history/archive for Bayesian hyperparameter optimization methods such as Tree Parzen Estimators (TPE) and Adaptive TPE (APTE) to generate new candidates. For example, TPE uses this initial knowledge to build a surrogate and generate new configuration candidates via Expected Improvement."
This is not very specific, what are the exact hyperparameter configurations that are being utilized? Important details seem to be missing here.
- Details are also missing for the LS Computation: which hidden states are being used? What is f respectively for LSTM and RHN, how is J obtained respectively? What was your choice of T and K?
2.) No code is provided, which adds to the irreproducibility.
3.) No theory is provided and it doesn't become clear from the submission why the LS-based metric is working. There is an heuristic explanation in figure 1, but the claim that "While early perplexity is not indicative of the estimated accuracy, LS-based distance on the other hand appears to be consistent [...]", seems only anecdotal. It would be great to show that more systematic, e.g., a scatter plot of early vs late perplexity vs a scatter plot of LS-distance vs late perplexity and the corresponding correlations.

4.) The paper claims that the hyperpruning methods would beat the state of the art, but the architectures being used (LSTM and LSH networks) seem not to be state of the art for language modeling, it would be interesting to see how LSH would perform on actual state of the art models. Moreover, the PTB benchmark is a very small data set, so it is unclear if the hyperpruning method would still perform on par when used on more recent benchmarks and architectures. Of course, the result is still of academic interest.
5.) The submission doesn't discuss the computational complexity. LS calculation involves QR-decomposion (which is cubic in Jacobian size), and also matrix-matrix multiplication (also cubic), so if I am not mistaken, this should become more expensive for larger problems, correct? Would the computational advantage still persist for state-of-the-art language models?

Minor issues
* Figure 1 A: x-axes labels/ticks are missing, is that just an illustration or actual LS curves?
* Figure 1B:
* How does the 2D LS-space look like? It would be good to add a (supplement) figure to plot embeddings in the LS space.
* Why is there an increase o the perplexity from the pruning ration of 0.6 to 0.5 in figure 3?
* It would be good to plot the test perplexity as a function of epochs
* Not clear what hyperparameters are actually used
* baseline fair? Maybe compare to something else than LS-based metric.
* Why increase perplexity from a pruning ration of 0.6 to 0.5?
* Shouldn't time efficiency (figure 3 right) be also compared with selfish RNN?
* (This might also just be a misunderstanding on my side.) It seems that the optimization of e_i and e_0 was only done for the LS-based metric and not for the loss-based metric. I was wondering what would happen if e_i and e_0 were optimized for the loss-based metric instead and then tested on the LS-based metric.


**Summary Of The Paper:**

The paper proposes a novel "Hyperpruning" method for RNNs, based on the Lyapunov spectrum. The idea is to select good pruned candidate networks after a few training steps based on their distance in the 'Lyapunov spectrum space' from the unpruned network.
The paper claims to achieve with this method state-of-the-art accuracy on the PTB benchmark with stacked LSTM and RHN,  while using fewer training epochs.


**Summary Of The Review:**


The submission suggests an innovative and novel hyperpruning method for RNNs based on Lyapunov exponents after a few training steps, resulting in higher performance and reduced computational cost.
The main issue of the submission are a lack of a theoretical understanding of how and why it works (including limitations/when it would break down) and a lack of scalability to real-world problems.

---

> ### Author Response · Authors · 2022-11-16
> **Response to the Reviewer mZ4i**
>
> **Response**
>
> We thank the reviewer for a thoughtful review and valuable feedback. We have addressed the comments raised by the reviewer and revised the manuscript accordingly. We describe the revisions and provide point-by-point clarifications below.
>
> **Strength and Weaknesses:**
>
> **W1. Variants generation:** We thank the reviewer for asking the missing details in the variants generation section. Basically, the remaining n candidates in T are used as the first n trials for TPE or ATPE. For example, if we need to generate n/2 new trials, we will set the number of trials for TPE or ATPE to (n + n/2). Then for the first n trails, we assign every candidate in T. For the remaining n/2 trails, TPE and ATPE will generate them based on all previous trials. Since those n candidates have already been trained, we can directly assign the values, i.e., LS-distance to the trials without training them again.
>
> **W2. LS Computation:** We thank the reviewer for asking for further information about LS computation. 1) We use all the hidden states in Stacked LSTM and RHN for the LS computation. For example, in Stacked LSTM, there are two layers and each layer has 1500 hidden states, so the number of hidden states used for LS computation in Stacked LSTM is 3000. For RHN, it has one layer with 830 hidden states, so we use all 830 hidden states for LS computation; 2) Function f for LSTM and RHN is their forward propagation function. We include those forward propagation functions in the Supplementary Matrial section B;  3) We derive the analytical expression of the Jacobian of both LSTM and RHN. We include the derivation of the Jacobian in the Supplementary Material section B; 4) T is the length of the sequence which is the same as the length of backpropagation through time. We use T=35 in both Stacked LSTM and RHN experiments. For K, we use 2 for both cases. As shown from Table. 4 in the manuscript, the difference between using 2 and 10 samples does not change the averaged LS significantly. Also, since LS computation is computationally expensive and memory heavy, a small K can prevent the algorithm from computational overhead.
>
> **W3. Code:** We provide the code in the attached zip file. People can follow the ReadMe file to reproduce the results in this paper. The code will be available in Github after the publication of the paper.
>
> **W4. Theory of LS-based metric:** We thank the reviewer for suggesting to include further theoretical explanation and a systematic comparison between early perplexity and LS-based distance.
> In terms of the theoretical underpinnings for LS for RNN, these were introduced in [1] and since our work is focusing on the application of LS we did not repeat these or elaborated more on these due to lack of space in the main mansucript. As suggested by the reviewer, we will include these and further clarifications in the supplementary materials. Also, as suggested b the reviewer, we now include a scatter plot of early LS-based distance vs. late perplexity and early perplexity vs. late perplexity in Fig.1 of Supplementary Material section C. We show the quantitative comparison using Pearson Coefficient which evaluates to what degree a monotonic function fits the relationship between two random variables, the higher the better. It shows that early LS-based distance has much higher Pearson Coefficient than early perplexity at the beginning of training (epoch 3: 0.68 vs. 0.45) and at early training (epoch 15: 0.73 vs. 0.27). These indicate that LS-based distance is informative and predictive in comparison in early stages of training than perplexity.
>
> **W5. Architecture and Dataset generalization:** We thank the reviewer for suggesting performing LSH on other SOTA language modeling networks. Indeed, one of prospective directions would be to apply LSH to Transformers which are widely used SOTA language model. This is beyond the scope of this work since in this work we wanted to establish that LSH is able to achieve better accuracy than other pruning methods by hyperparameter search. The experiments we did on Stacked LSTM and RHN trained with Penn Tree Bank dataset are commonly used as benchmarks in RNN pruning literature [2]. Notably, there are no special constraints on our approach to those two architectures or PTB dataset and therefore this approach is expected to be generalizable to other RNN models or datasets.

---

> > ### Author Response · Authors · 2022-11-16
> > **Response to the Reviewer mZ4i (Ctd1)**
> >
> > **W6. Computational complexity:** We thank the reviewer for pointing out the computational complexity of Lyapunov Spectrum (LS) computation. It is correct that LS computation could be computationally expensive and memory heavy since it involves QR decomposition and Jacobian matrix multiplication in comparison to computing the perplexity for example. In our experiments, we use an NVIDIA GeFORCE RTX 2080 Ti GPU and it takes about 6s to compute the LS computation for each sample. However, as we mention in Table. 4 of the manuscript, the difference between using 2 and 10 samples is not significant, so in our experiments, we just use two samples for the LS computation which takes about 12s. Compared with the time of training an epoch (90s), LS computation is only a fraction of this time and slows down candidate selection process by 13% compared to perplexity-based one. Note that the extensive training process does not require LS computation. Given the effectiveness of LSH in choosing more accurate candidates, this efficiency tradeoff is reasonable since saves hours of extensive training as we show in Figure 3 (Right) in the manuscript. Moreover, for larger problems, even if the LS computation is expensive since only a few samples are required, the computation overhead resulted by the LS computation is still acceptable compared to the whole training. We include this additional discussion in the Supplementary Material section D.
> >
> > **Minor Issues:**
> >
> > **M1. Figure 1 A x-axes:** We thank the review for pointing out the confusion about the x-axis label. The x-axis labels are training stage, and the ticks are stages at pre-training, epoch 1, 3 and 5. For those curves in Fig.1, all of them are obtained from experiments.
> >
> > **M2. Figure B 2D LS-space:** The LS-space of a pruned model and dense model is shown in Fig. 2 of Supplementary Material section F.
> >
> > **M3. Perplexity Increase:** We thank the reviewer for noticing this unexpected increase of perplexity in Fig. 3 of the manusript. This same increase of perplexity was also observed in previous work [2]. Intuitively, smaller pruning ratio leads to a better performance (lower perplexity) since more weights are available for training. However, pruning acts as an extra regularization term for the network optimization and it turns out that pruning ratio of 0.6 serves as a better regularization. This can also explain why the pruned model outperformed the dense model as shown in Fig. 3 (Left) of the manuscript.
> >
> > **M4. Test perplexity plot:** The plot of epoch vs. test perplexity of the model selected by LSH is shown in Fig. 2 (Right) of the Supplementary Material section F.
> >
> > **M5. Hyperparameter choice:** We thank the reviewer for pointing out the missing hyperparameter used during training. We show the hyperparameter of Stacked LSTM and RHN in Table. 1 and include them in the Supplementary Material section A.
> >
> > Table 1: Hyperparameters list: dimension of hidden units (H-dim), number of layers (Layers), dimension of input embedding (Emb), Optimization (Opt), Learning rate (LR), Non-monotone interval for SNT-ASGD (Non-mono), Training batch size (BS), Back propagation through time (BPTT), Gradient Clip (Clip), LS computation batch size (LS-BS), Sparsity decay schedule (Decay Sche), Encoder and Decoder weight tied (Tied).
> >
> > | Model | H-dim | Layers | Emb | Opt | LR | Non-mono | BS |
> > |:---:|:---:|:---:|:---:|:---:|:---:|:---:|:---:|
> > | Stacked LSTM | 1500 | 2 | 1500 | SNT-ASGD | 40 | 5 | 20 |
> > | RHN | 830 | 1 | 830 | SNT-ASGD | 15 | 5 | 20 |
> > |  |  |  |  |  |  |  |  |
> > | Model | BPTT | Dropout | Epochs | Clip | LS-BS | Decay Sche | Tied |
> > | Stacked LSTM | 35 | 0.65 | 100 | 0.25 | 2 | Cosine | False |
> > | RHN | 35 | 0.65 | 500 | 0.25 | 2 | Cosine | True |
> >
> > **M6. Other baseline:** We use loss-based metric as the baseline and compare it with our LS-based metric since without any other knowledge, loss is the intuitive metric for model selection.
> >
> > **M7. Time Efficiency comparison:** We thank the reviewer for suggesting a time efficiency comparison between LSH and selfish RNN. However, Selfish RNN does not involve any hyperparameter search and thus is not applicable for time efficiency comparison. Therefore, instead of comparing the time efficiency with Selfish RNN, we compare hyperparameter configuration selected by LSH with Selfish RNN and show that LSH is able to find better configurations by hyperparamter search instead of prososing new architectures or algorithms.

---

> > > ### Author Response · Authors · 2022-11-16
> > > **Response to the Reviewer mZ4i (Ctd2)**
> > >
> > > **M8. e_0 and e_i optimization for loss-based:** From previous observations of the poor correlations between early perplexity and final perflexity, e_0 needs to be very large to achieve reasonable correlations. To verify this, we conduct an experiment on Stacked LSTM where the death rate is fixed and only 24 candidates with different methodological hyperparameter configurations are considered. It shows that the best configuration obtains lowest perplexity among all candidates after 75 epochs of training. This is about ¾ of the whole training (75 out of 100 epochs) and does not accelerate the candidate selection process too much compared to the baseline loss-based full training search.
> > >
> > > [1] Ryan Vogt, Maximilian Puelma Touzel, Eli Shlizerman, and Guillaume Lajoie. On lyapunov exponents for rnns: Understanding information propagation using dynamical systems tools. Frontiers in Applied Mathematics and Statistics, 8, 2022a. ISSN 2297-4687. doi: 10.3389/fams.2022.818799.
> > > URL https://www.frontiersin.org/article/10.3389/fams.2022.818799
> > >
> > > [2] Shiwei Liu, Decebal Constantin Mocanu, Yulong Pei, and Mykola Pechenizkiy. Selfish sparse rnn training. In International Conference on Machine Learning, pp. 6893–6904. PMLR, 2021.

---

### Decision · Program_Chairs · 2023-01-20

**Decision:**

Reject

**Justification For Why Not Higher Score:**

The problem of "hyperpruning" is a good one, but the proposed method for solving it and the evaluation are insufficient. A good question is insufficient for publication alone without (at least) a simple, solid, reproducible baseline for others to build on.

**Justification For Why Not Lower Score:**

N/A

**Metareview: Summary, Strengths And Weaknesses:**

**Summary:** This paper proposes both a new research problem and a solution to that problem. Thew new problem is *hyperpruning* - that there are a huge variety of pruning techniques in the literature (and each pruning technique has lots of hyperparameters); determining which technique to use in which circumstance is a nightmare. I work in pruning myself, and I agree that this is a challenge and an interesting problem. The authors look into solving this problem in a style similar to hyperparameter search (although with slightly different knowledge/constraints, as the authors acknowledge).

**Strengths:** This is an interesting new problem, and one of the most valuable possible contributions that a paper can make is to propose a new problem in the literature.

**Weaknesses:**
* Strange choice of settings. The pruning literature typically focuses on computer vision (usually image classification for ResNets and vision transformers) or NLP using transformers. I won't argue that this is necessarily the right choice, but it is the basis for comparison with prior work. This paper is incomparable to that work.
* Small-scale settings. The paper does NLP work, but focuses on RNNs on PTB. RNNs are not a modern setting, and PTB is a very small-scale setting. Over the years, the pruning community has learned the hard lesson that results in niche and small-scale settings don't tend to generalize very well, and - at this point - the bar for publication is to look at some subset of ResNets on ImageNet, transformers for masked language modeling (e.g., BERT), autoregressive language modeling, translation, etc. These small-scale settings may not generalize, and the lyapunov metric may indeed become very expensive to compute on bigger networks when larger amounts of data may be required.

**Other comments:**
* The choice of the lyapunov metric is a bit of an odd one. As far as I can tell, it looks at the underlying representation learned by the network as a means for comparing networks, which seems unreasonably restrictive. It's possible that two equally good networks may have very different representations - why would you want to rule out that possibility?
* There was some confusion over how new networks are generated during the search process. This is something that needs a significant amount of attention: there are a lot of possible subnetworks out there, and the promise of also picking among many pruning methods (as stated in the introduction) is especially tricky. This needs to be addressed in more detail. This is also where having a wide variety of different networks/datasets/tasks would be helpful: as framed in this paper, a good hyperpruning method should produce great results across a wide variety of different settings with minimal overhead, whereas any specific pruning method/hyperparameter choices should fall short in at least some settings. This is another reason the current evaluation is insufficient.
* I disregarded the review from reviewer cX2r - the review was brief and had no serious technical content.

**Recommendation:** For the above reasons, I recommend rejection.

**Summary Of Ac-Reviewer Meeting:**

N/A